# A Theoretical Framework for Escaping Local Optima in MSE toward Global Convergence

## Abstract

Deep learning models are trained by minimizing loss functions such as mean squared error (MSE) or cross-entropy, but these objectives are highly non-convex. As a result, optimization often encounters local optima, saddle points, or sharp valleys that hinder convergence and generalization. Although many heuristic approaches, such as momentum, Adam, help mitigate these issues, they provide limited theoretical understanding. In this work, we present a theoretical study of the optimization of MSE. We first provide a mathematical characterization of local optima under MSE and contrast them with those of cross-entropy, identifying when and how they arise. Building on this analysis, we introduce a modified optimization algorithm that explicitly accounts for these properties. Unlike heuristic methods, our approach offers theoretical guarantees for avoiding spurious local traps. Our experiments show that the proposed method reliably avoids local optima and converges more effectively than existing optimizers in MNIST, CIFAR10 and CIFAR100 with simple CNN. Our work provides both new insight into MSE optimization for training deep networks.

## 1 Introduction

Deep learning models are trained by minimizing a loss function, which measures how far the model's predictions are from the ground truth(Hinton et al. (2012)). However, commonly used loss functions such as cross-entropy(Mannor et al. (2005); Jamin & Humeau-Heurtier (2019)) or mean squrared error (MSE)(Zhou et al. (2022)) – are non-convex due to the model's activations(Jain et al. (2017)). This means they do not have only a single global optimum. Instead, they have complex landscape with many local optima such as saddle points(Choromanska et al. (2015)). Because of this, many studies(Cheridito et al. (2024); Nguyen (2021)) have focused on overcome the local optima where the optimization stops even though the model is not yet at the ground truth.

Over the years, researchers have tried many ways to reduce this problem(Hinton et al. (2012); Zhang et al. (2022)). One family of approaches works by changing how gradients are used in training process. For example, momentum(Sutskever et al. (2013); Jelassi & Li (2022)) makes the trend to go to the certain direction, and make it easier to escape small valleys. Adam(Kingma & Ba (2014)) further adapts the learning rate for each parameter, so that the optimizer can avoid local optima and train the model more stably. These methods have become extremely popular in deep learning because they are effective for various cases.

Another kind of works takes a different view: instead of only considering the gradients, they focus on understanding the shape of the loss landscape itself(Zhou et al. (2022)). For instance, researchers have shown that in very high-dimensional spaces, it is not local minima but saddle points that are the major challenge, because they appear more often(Dauphin et al. (2014)). They proposed the saddle-free Newton method, which uses curvature information from the Hessian to escape from saddle points. Later, some studies Agarwal et al. (2017a;b) developed a second-order algorithm that has theoretical guarantees for finding approximate local minima faster than conventional gradient descent.

Despite this progress, most existing methods are still heuristic(Tian & Fong (2016); Kaveh & Mesgari (2023)). In fact, there is still no simple, precise mathematical description of what the local optima of MSE actually appear. Interestingly, Reddi et al. (2019) shows that Adam can fail to converge, despite its widespread use. As a result, they often depend on heuristic strategies, such as

keeping the gradient moving artificially so the optimizer does not fall in local optima. These methods may work in some cases, but they sometimes cause lower performance in a few cases(Zhou et al. (2020)). Intuitively, if the global optimum is very sharp and narrow, then the momentum can simply pass by it without entering. This shows that escaping local optima is not a one-size-fits-all problem, but depends heavily on both the model architecture and the loss function.

In this paper, we focus on MSE loss and make two main contributions. First, we present a mathematical analysis of local optima in MSE compared to Cross-entropy, and what properties they have. Second, building on this analysis, we propose a new loss function to optimize while avoiding such local traps in a principled way. Unlike conventional heuristic approaches, our method provides clear guarantees and adapts naturally to the conventional training process with Pytorch.

By combining theoretical insight with practical optimization, our study aims to characterize local optima through optimality condition of objective, and proposes a loss function to modify the direction of optimization. This is especially important because it is grounded in a solid mathematical basis. The main implication is that if this method converges but the loss value is not zero, this is due not to local optima of the loss, but rather to the limited expressive power of the model.

## 2 OPTIMALITY CONDITION OF CRITERIA

In this section, we define criteria and differentiate the objective functions to derive the optimality conditions.

### 2.1 CROSS-ENTROPY LOSS

First, we consider the cross-entropy loss function, which is widely used in classification tasks:

$$L_{\text{CE}}(w) = \sum_{i=1}^{c} -y_i \log f_i(w). \tag{1}$$

By differentiating $L_{\text{CE}}(w)$ with respect to $w$, we obtain the optimality condition for cross-entropy:

$$\sum_{i=1}^{c} \frac{y_i}{f_i(w^*)} \nabla_w f_i(w^*) = 0. \tag{2}$$

If $y$ is encoded using one-hot encoding, then all labels are zero except for the true label. Therefore, it is not affected by the singularity of the Jacobian.

### 2.2 MEAN SQUARED ERROR (MSE) LOSS

Next, we begin by defining the Mean Squared Error (MSE) loss with respect to the model parameters $w \in \mathbb{R}^d$:

$$L_{MSE} = \frac{1}{2} \|y - f(w)\|^2, \tag{3}$$

where $y \in \mathbb{R}^c$ is the target vector and $f(w) \in \mathbb{R}^c$ is the model output.

To find the stationary point, we take the gradient of $L(w)$ with respect to $w$. The first-order optimality condition is given by

$$D_w f(w^*)^\top (y - f(w^*)) = 0, \tag{4}$$

where $Df(w^*) \in \mathbb{R}^{c \times d}$ denotes the Jacobian matrix of $f$ at $w^*$.

Equation (2) can be equivalently written as the summation over each output dimension:

$$\sum_{i=1}^{c} (y_i - f_i(w^*)) \nabla_w f_i(w^*) = 0. \tag{5}$$

This condition characterizes the optimality of the MSE loss. It is similar to (2), since both conditions represent a linear combination of class-wise gradients. According to optimality condition of MSE, the training is finished in following three cases.

1. $y_i = f_i$ for $i = 1 \cdots c$

2. $\nabla_w f_{i=1 \cdots c} = 0$

3. The linear combination of $\nabla_w f_{i=1 \cdots c}$ is zero

The first case corresponds to the global optimum because (3) is strictly greater than zero. The second and last cases should represent local optima of the MSE. The second case is minor because each gradient of activations are not zero, except for Relu. Even when using ReLU, the case where all gradients become zero is a special case and is not a primary consideration. On the other hand, the last case is more critical: once the parameter vector falls into the null space of the Jacobian matrix, the gradient of the loss becomes zero, and momentum cannot restore the gradient from the null space.

## 3 MODIFIED MSE

As we have seen, the MSE loss itself can lead to local optima, so we propose a modified MSE formulation to mitigate this issue. We refer to this modified loss as QMSE. In the next, we present some properties of QMSE.

### 3.1 QMSE

In Equation (4), local optima may arise due to the null space of the Jacobian. To address this, we modify the MSE by transforming the residual vector $(y - f)$. Specifically, if we use a quadratic norm instead of the standard 2-norm, the loss and the gradient of the loss becomes

$$L_{QMSE} = \tfrac{1}{2}\|y - f(w)\|_Q^2, \tag{6}$$

$$\nabla_w L_{QMSE} = -D_w f(w)^\top Q(y - f). \tag{7}$$

Now, we can handle the residual vector $(y - f)$ by $Q$ and should derive good $Q$ for $\nabla_w L_{QMSE}$.

To address this, we setc

$$Q = (y - f + v)(y - f + v)^T, \tag{8}$$

where $v$ is the random unit vector subject to $D_w f^T v \neq 0$. If any matrix has $xx^T$ shape, then the matrix has only one eigenvalue, $\|x\|_2^2$. And the corresponding eigenvector is $x$. So $Q$ has an eigenvalue, $\|y - f + v\|_2^2$ and the corresponding eigenvector, $(y - f + v)$. Although it is possible to find an analytic vector $v$ that avoids the null space of the Jacobian. However, explicitly considering the Jacobian needs large computational cost. A more practical approach is to use random sampling, and if the sampled vector still lies in the null space, simply resample.

### 3.2 PROPERTIES

In this section, we show some properties of QMSE using the reformulation in (7). First, we plug in $Q$ from (8) into (7) and simplify the expression. Then, (7) becomes

$$\nabla_w L_{\text{QMSE}} = D_w f^\top v \left( \|y - f\|_2^2 + \langle v, y - f \rangle \right). \tag{9}$$

$\langle a, b \rangle$ denotes the dot product between $a$ and $b$. The direction of training is determined by the Jacobian $D_w f$ and $v$, while the other terms inside the parentheses determine the magnitude of the vector(they are a scalar value).

We list some basic properties of QMSE in some cases at local optima with respect to $v$.

- *Near the global optimum*, $\|y - f\|_2^2$ is small, and the transformed residual vector $Q(y - f)$ can escape the null space more carefully. In the opposite case, it should escape more aggressively.

- *When $v$ and $(y - f)$ are almost parallel*, $\langle v, y - f \rangle$ is large, and the transformed residual vector $Q(y - f)$ can escape the null space more aggressively. In the opposite case, it should escape more carefully.

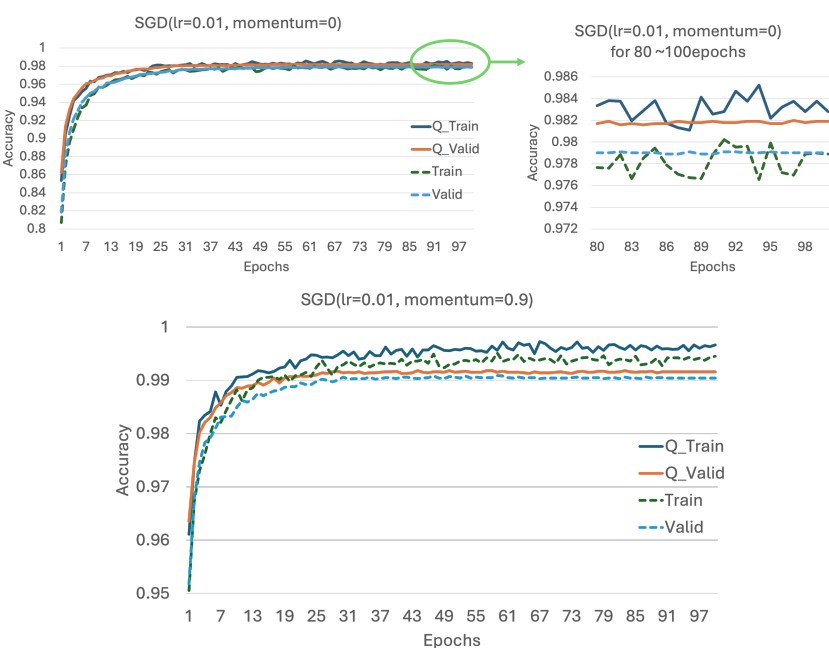

Figure 1: Training and test accuracy of the simple CNN on the MNIST dataset. Our method is denoted as Q_Train and Q_Valid, while the others are trained with conventional MSE. The upper figure shows the four accuracy curves obtained using SGD with zero momentum, while the bottom figure shows the results with a momentum of 0.9. The top-right plot provides a zoomed view of the accuracy between epochs 80 and 100. In all MNIST experiments, the learning rate was set to 0.01 and 100 epochs.

## 4 EXPERIMENT

In this section, we aim to verify whether our method achieves better performance than standard methods by reaching the global optimum in train set. We employ a simple CNN consisting of two convolutional layers followed by two fully connected layers. Using the MNIST, CIFAR-10, and CIFAR-100 datasets, we evaluate how well our method performs under various datasets and optimizer settings, based on both training and test accuracy. In particular, we compare our method with momentum. We do not use the loss for comparison, since QMSE is based on the $Q$-norm while MSE is based on the $\ell_2$-norm. This makes both numerical values not directly comparable. However, due to the optimality conditions, both losses share the same local and global optima. Therefore, we use accuracy as the evaluation metric.

### 4.1 MNIST

We evaluate our method on the MNIST dataset. Figure 1 shows the training and test accuracy of QMSE and MSE. The learning rate is set to 0.01 due to the rapid convergence speed. QMSE and MSE are achieved nearly 1.0 accuracy. However, based on the training accuracy, QMSE converges to a lower optimal point in loss, so we can find that it outperforms in both training and validation accuracy. When we use QMSE with momentum, it achieves the highest performance on MNIST in our experiments.

### 4.2 CIFAR-10

Our goal is to overcome the local optima of MSE, and this is supported by the training accuracy. On CIFAR-10 dataset, QMSE and MSE achieve similar validation accuracy, while train accuracy shows a difference. In the right panel of Figure 2, where SGD is used with momentum, both methods achieve nearly 1.0 training accuracy. In contrast, when SGD has no momentum, QMSE shows

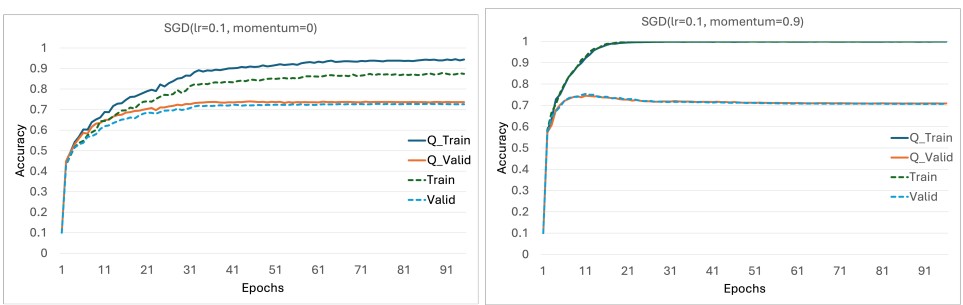

Figure 2: Training experiments of the simple CNN on the CIFAR-10 dataset. The upper panel shows the accuracy curves obtained using SGD with zero momentum, and the lower panel shows the results with a momentum of 0.9. For all CIFAR-10 experiments, we set the learning rate to 0.1 and trained for 100 epochs.

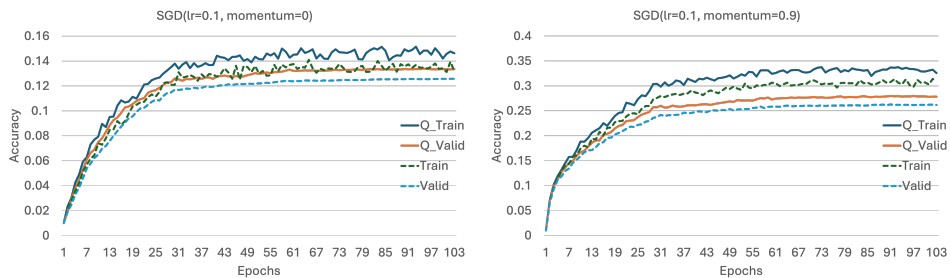

Figure 3: Training results of the simple CNN on the CIFAR-100 dataset. The upper panel shows the accuracy curves obtained using SGD with zero momentum, and the lower shows the results with a momentum of 0.9. In all CIFAR-100 experiments, the learning rate was set to 0.1, and training was performed for 100 epochs.

significantly higher training accuracy. These results indicate that QMSE, combined with momentum or not, helps overcome local optima in the loss landscape on the training dataset.

## 4.3 CIFAR-100

On the CIFAR-100 dataset, momentum significantly affects accuracy, whereas QMSE impacts sensitivity. It shows in Figure 3 Training with SGD without momentum achieves nearly 0.15 training accuracy, while using momentum increases it to almost 0.35. First, the model is too simple to effectively train on this dataset. It results in overall low accuracy. Additionally, CIFAR-100 has 100 output channels, which makes the loss landscape more complex and introduces more local optima. Nevertheless, QMSE improves performance on complex datasets, with or without momentum.

## 5 CONCLUSION

In this work. We propose QMSE, a modification of MSE to overcome local optima in the loss landscape. Through experiments on MNIST, CIFAR-10, and CIFAR-100 datasets, we show that QMSE improves performance, and it can make more improvements when combined with momentum. While both QMSE and MSE achieve similar validation accuracy on CIFAR-10, QMSE exhibits outperforms on train accuracy on all of three datasets. These results highlight that QMSE provides a robust optimization strategy that enhances optimizer's ability to reach better optima. Moreover, high performance in train dataset does not necessarily indicate good generalization. Future work will explore the integration of QMSE with more advanced architectures and various optimization strategies to further improve performance on large-scale and complex datasets.

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

# A APPENDIX

We only use a Large Language Model (LLM) to assist in translating Korean into English.

