# OpenReview forum: "A Theoretical Framework for Escaping Local Optima in MSE toward Global Convergence"
_ICLR.cc/2026/Conference — Submitted to ICLR 2026_

### Official Review · Reviewer_uHc4 · 2025-10-29

**Soundness:** 1
**Presentation:** 1
**Contribution:** 2
**Rating:** 0
**Confidence:** 4

**Summary:**

This paper is a theoretical study of the optimization landscapes of loss functions like MSE and cross-entropy. The authors present a new loss based on known claims in the field and show experimental results to support the claim that their new loss function avoids local optima and converges more effectively than existing optimizers on a specific set of datasets.

**Strengths:**

I believe authors have attempted to study a interesting problem of escaping local optima and analyzing existing optimizers characteristics. I feel the authors have an interesting idea but the whole presentation lacks rigor and is confusing with many errors.

**Weaknesses:**

This paper is definitely not at publishable quality right now.

1) Lot of grammatical errors, QMSE - only used as abbreviation no concrete explanation.
2) Missing related work.
3) I also find many logical flaws in the motivation and general study here. The issues identified are really about the neural network optimization landscape, not MSE specifically. I am not sure if the authors are confused between ideas of optimization surface and loss function. Ex: The paper states MSE loss is 'non-convex due to the model's activations', but then they say that the issue is in with the MSE?
4) Additionally, central claim that MSE has inherent local optima problems is not well-supported.
5) The supporting mathematical claims has errors and there is lack of detail on QMSE.

The issues discussed are really about the neural network optimization landscape (the Jacobian structure, linear dependencies in gradients), not something inherent to MSE specifically. I am not sure where are the authors are going with the claims. What is $f$ here? How is it defined?

**Questions:**

NA

---

### Official Review · Reviewer_ebWV · 2025-10-31

**Soundness:** 1
**Presentation:** 2
**Contribution:** 1
**Rating:** 2
**Confidence:** 3

**Summary:**

This paper addresses the local optima problem in the optimization of MSE loss in deep learning. It first provides a mathematical characterization of local optima for MSE (compared with cross-entropy) and identifies their root cause (the null space of the Jacobian matrix leading to zero linear combination of gradients). Building on this analysis, the authors propose QMSE, a modified MSE loss. Experiments on MNIST, CIFAR-10, and CIFAR-100 with a simple CNN show that QMSE outperforms vanilla MSE in training accuracy.

**Strengths:**

QMSE is a lightweight modification of MSE, requiring no major changes to the training pipeline.

**Weaknesses:**

- Most theoretical analyses are straightforward and largely well-known, and some descriptions lack rigor (e.g., Line 157).

- Experimentally, QMSE improves training accuracy but achieves similar validation accuracy compared to vanilla MSE (e.g., Fig. 2(a)).

- Both theoretical analysis and experimental results are insufficient to support this method.

**Questions:**

See Weaknesses.

---

### Official Review · Reviewer_tuQd · 2025-10-31

**Soundness:** 1
**Presentation:** 3
**Contribution:** 2
**Rating:** 2
**Confidence:** 4

**Summary:**

The goal is to provide a theoretical framework for escaping local optima in MSE toward global convergence. This is a well motivated problem as non-linear optimization on MSE loss are generally notoriously hard to optimize. The authors address this through standard analysis of MSE and Cross-entropy losses. The main contribution is the presentation of a QMSE, a modified MSE loss that uses a quadratic norm instead of the standard L2 norm. Standard analysis of this loss supports some beneficial properties for optimization. Numerical investigation training CNNs on MNIST, CIFAR-10, and CIFAR-100 indicate modest to negligible practical impact on training and validation loss.
The submission is generally clear, though several writing issues are present. Several variables are not defined, some equations are mis-referenced. In particular, I found the text associated with Section 3.1, which presents the QMSE loss, to be difficult to read. Some claims are simply not supported by the presented results (e.g., QMSE has enhanced sensitivity).

**Strengths:**

The introduction of QMSE is potentially impactful and is the main contribution.

**Weaknesses:**

There is no formal analysis of the optimization dynamics of the error functions
There is no numerical analysis of the critical points of the error functions
The demonstrated practical effects of the proposed loss are negligible
Only small models are presented.
The references are extremely spars
The writing needs to be clarified in many places.
Some of the observations about MSE loss have been made before (e.g., Frye et al., Neural Computation).
Not enough detail is provided on the numerical experiments to enable reproduction
No software is provided.

**Questions:**

n/a

---

### Official Review · Reviewer_vo6u · 2025-10-31

**Soundness:** 1
**Presentation:** 1
**Contribution:** 1
**Rating:** 2
**Confidence:** 4

**Summary:**

The paper studies optimization with MSE in deep networks. It derives a first-order stationarity condition for MSE and argues that spurious local optima can occur when the parameter vector falls into the null space of the Jacobian. To mitigate this, it proposes a “QMSE” objective that replaces the usual ℓ2 norm with a data-dependent quadratic form $Q=(y-f+v)(y-f+v)^T$, where v is is a random unit vector chosen so that $J^Tv\neq 0$. Experiments on MNIST, CIFAR-10, and CIFAR-100 with a simple CNN claim that QMSE attains higher training accuracy (and sometimes validation accuracy) than vanilla MSE, especially without momentum.

**Strengths:**

1. A simple, implementable idea (injecting a random direction v) that—at least heuristically—ensures a non-zero projected gradient and can be realized via efficient VJP/JVP operations.

**Weaknesses:**

1. “Theoretical guarantees” are claimed but not provided. The abstract says the method “offers theoretical guarantees for avoiding spurious local traps,” but the body contains no formal theorems, assumptions, or proofs establishing convergence or even non-vanishing gradients beyond heuristics.
2. Method novelty and positioning. The proposal effectively adds a random directional component to the residual so that $J^Tv\neq0$, which resembles perturbed/annealed gradients or gradient noise injection, widely studied for escaping saddles. The paper does not position QMSE relative to these families of methods nor to modern sharpness-aware or entropy-based objectives. This raises novelty concerns.
3. Limited scope and baselines. Only a “simple CNN” and small datasets are used; no modern architectures (e.g., ResNet-18) or larger scales. Comparisons exclude common baselines that also encourage escape from sharp/degenerate regions (Adam, RMSProp, SGD+noise, Lookahead, SAM, label smoothing, CE).
4. Focus on training accuracy with weak generalization gains. The paper itself notes validation accuracy is similar to MSE on CIFAR-10, with the main gains in training accuracy; on CIFAR-100 the model underfits badly (train acc ~0.15–0.35). Thus, evidence that QMSE improves useful convergence (i.e., generalization) is limited.
5. Clarity & writing. Numerous grammatical issues and ambiguous phrasing (e.g., “mean squrared,” “modi-fied,” “keep the gradient moving artificially”) and inconsistent notation formatting. This affects readability and precision. (Examples across the Abstract, Intro, and Sections 2–3.)

**Questions:**

1. The delineation between global and local minima is too coarse and offers limited insight. Prior work [1] analyzes MSE in an unconstrained feature model and proves there are no spurious saddle points—every local minimum is global. This suggests your Cases 2 and 3 effectively collapse to Case 1. Do you contend that [1] is incorrect, or do your assumptions differ in a way that invalidates its result? Please clarify and provide a formal basis (e.g., a counterexample, differing setting, or additional constraints and corresponding proof to support your claim).

[1] Zhou et al. On the optimization landscape of neural collapse under mse loss:Global optimality with unconstrained features.

---

### Meta-Review · Area_Chair_ZSh1 · 2025-12-24

**Summary:**

The reviewers unanimously agree that the soundness of this work is poor. In particular, the abstract and introduction state that the method "offers theoretical guarantees for avoiding spurious local traps," but the body of the paper contains no formal theorems, assumptions, or proofs establishing convergence or even non-vanishing gradients beyond heuristics. Some reviewers also complained about logical flaws in the motivation and grammatical errors throughout the paper. The majority of the reviewers further pointed out that details of the experimental setup are missing and/or that the messages drawn from the experiments may not be fully justified.

**Reviewer Concerns:**

This question is not applicable, since the authors did not participate in the rebuttal.

**Reviewer Scores:**

This question is not applicable, since the authors did not submit the rebuttal.

---

### Decision · Program_Chairs · 2026-01-26

Reject